# Impact of Maternal Metabolic Status on Human Milk Oligosaccharide Composition: A Population-Based Cross-Sectional Study in Central South China

**DOI:** 10.3390/nu17091480

**Published:** 2025-04-28

**Authors:** Zhi Huang, Shurong Luo, Yuxin Li, Ziming Li, Chuanzhu Yi, Yan Zhang, Yuming Hu, Bo Chen

**Affiliations:** 1School of Medical Laboratory, Hunan University of Medicine, Jinxi Road No. 492, Huaihua 418000, China; hhsfbyzhige@hotmail.com (Z.H.); 19015941646@163.com (Y.Z.); 2School of Chemistry & Chemical Engineering, Hunan Normal University, Lu Mountain Road No. 286, Changsha 410081, China; luoshurongco@126.com; 3Hunan Institute for Drug Control, Jinxing Middle Road No. 469, Yuelu District, Changsha 410001, China; 15974176296@163.com; 4The Department of Toxicology, Hunan Provincial Center for Disease Control and Prevention, Xinglian Road No. 861, Laodaohe Street, Kaifu District, Changsha 410005, China; hncdclzm@163.com (Z.L.); ycz1974@163.com (C.Y.)

**Keywords:** metabolic factors, lactating mothers, human milk oligosaccharides

## Abstract

**Background:** Human milk oligosaccharides (HMOs) serve as critical bioactive components supporting infant growth and development. However, the influence of maternal metabolic factors during lactation on HMOs remains to be fully elucidated. This study aimed to investigate the association between maternal metabolic factors and HMOs, as well as the potential mediating effects of these factors. **Methods:** An observational cross-sectional study was conducted in Central South China, enrolling 196 lactating mothers. HMOs were quantified using liquid chromatography-tandem mass spectrometry. Maternal metabolic factors were assessed through physical examinations. Associations between metabolic factors and HMOs were analyzed using linear regression, and mediation effects were evaluated. **Results:** HMOs from Central South China were predominantly composed of neutral fucosylated HMOs. Significant differences were observed in the levels of several HMOs across maternal age groups and lactation periods. The concentration of 3′-sialyllactose (3′-SL) exhibited a negative association with the pre-pregnancy body mass index (BMI) (β = −0.16, 95% CI: −0.29, −0.03; *p* = 0.02), while a positive association was found with maternal heart rate (β = 0.14, 95% CI: 0.01, 0.27; *p* = 0.04). However, these associations were different between secretor and non-secretor mothers. Associations of 3′-SL with pre-pregnancy BMI and maternal HR were only found in the secretor mothers. Triglycerides and low-density lipoprotein cholesterol mediated the associations between maternal pre-pregnancy BMI and 3′-sialyllactose (3′-SL). **Conclusions:** The variations of several HMOs among mothers from Central South China were associated with maternal age and lactation period. The concentration of 3′-SL was negatively correlated with maternal pre-pregnancy BMI. The potential mechanism underlying the influence of maternal BMI on 3′-SL levels may involve maternal lipid metabolism and genetic factors.

## 1. Introduction

Breast milk is the most natural and optimal source of infant nutrition. Exclusive breastfeeding is recommended during the first six months of life, followed by continued breastfeeding for up to two years or beyond [1]. Human milk oligosaccharides (HMOs), the third most abundant solid component in breast milk [2], play a crucial role in infant growth and development. Their benefits include modulating intestinal microbiota [3], exerting immunomodulatory effects [4], providing anti-infective protection [5], and promoting neurodevelopment [6]. Due to these functions, HMOs are now added to infant formula to improve health [7].

Multiple studies have reported significant variations in HMO composition across individuals and populations, influenced by diverse factors. Genetic determinants, particularly fucosyltransferase 2 (FUT2) and fucosyltransferase 3 (FUT3), play a central role in this variability by determining maternal secretor status (Se) [8]. Additionally, extensive research has shown that HMO composition evolves throughout the course of lactation [9,10]. Other influencing factors, including maternal age, ethnicity, and parity, were also found to be associated with the composition of HMOs. However, many prior reviews have overlooked critical details, and notably, not all of these factors are modifiable.

In recent years, maternal body mass index (BMI) has emerged as a modifiable factor influencing HMO concentrations, attracting significant research attention. Several studies have reported an inverse relationship between maternal pre-pregnancy BMI and the abundance of most oligosaccharides [11,12]. However, other studies failed to find an association between BMI and HMO composition [13]. Clarifying this discrepancy is crucial for developing targeted interventions to optimize breast milk composition. Maternal overweight and obesity during pregnancy represent significant metabolic risk factors, predisposing women to various metabolic derangements, including dyslipidemia, gestational diabetes [14], and preeclampsia [15]. Schönbacher et al. further suggested that serum oligosaccharides may be influenced by maternal metabolic factors, particularly lipid metabolism and BMI [16]. Nevertheless, the precise impact of maternal metabolic factors during lactation on HMO profiles in breast milk requires further investigation.

In this study, we aimed to assess individual-level variations in eight HMOs in a Central South Chinese population. Furthermore, we investigated potential associations and mediating effects of maternal metabolic factors on HMO profiles.

## 2. Materials and Methods

### 2.1. Study Design and Participants

This cross-sectional observational study was conducted in Xiangtan City, located in Central South China. Lactating women who delivered at the local maternal and child healthcare hospital were enrolled between September 2022 and February 2023. All participants met the following inclusion criteria: (1) maternal age between 18 and 40 years; (2) lactation period ranging from 3 to 24 weeks. Exclusion criteria included: (1) lactating mothers with mastitis; infectious diseases; chronic cardiovascular, hepatic, or renal diseases; neurological disorders; cancer; or a history of medication use affecting nutrient metabolism. A total of 196 women were recruited. The study protocol was approved by the Ethics Committee of the Hunan Provincial Center for Disease Control and Prevention (HNCDC-BJ2022005 and 10 October 2022). All participants provided written informed consent.

### 2.2. HMO Analysis

Breast milk samples were collected from all participants during interviews. Using manual breast pumps, trained staff collected 10–20 mL aliquots of milk between 8:00 and 10:00 a.m. following a standardized protocol. Each sample was immediately labeled with a unique identification number and stored at −80 °C until HMO analysis.

The liquid chromatography-tandem mass spectrometry (LC-MS/MS) method was established to quantify eight HMOs in breast milk, based on previously published methodologies [17,18,19]. The analyzed HMOs included: 2′-fucosyllactose (2′-FL), 3-fucosyllactose (3-FL), lacto-N-tetracose (LNT), lacto-N-neotetraose (LNnT), *N*-fucopentaose I (LNFP-I), lacto-N-difucosylhexaose II (LNDFH-II), 3′-sialyllactose (3′-SL), and sialyllacto-N-tetraose a (LSTa).

A 100 μL breast milk sample was centrifuged at 10,000 rpm for 15 min at 4 °C after thawing. Then, 50 μL of the lower aqueous phase was transferred to a new microtube and mixed with 100 μL of anhydrous ethanol. The mixture was incubated at room temperature for 30 min to allow precipitation, followed by centrifugation at 12,000 rpm for 10 min at 4 °C. After centrifugation, 50 μL of the supernatant was collected and diluted with ultrapure water. To enhance the resolution of the HMO isomers, a fluorescent derivatization step using 2-naphthylamine was employed to increase the hydrophobicity of the HMOs. Specifically, 50 μL of 0.5 M 2-naphthylamine and 50 μL of 1 M 2-methylpyridine borane were added to 100 μL of the diluted sample, followed by incubation at 50 °C for 1 h. Subsequently, 600 μL of chloroform was added, and the mixture was extracted twice. Finally, the upper aqueous phase was filtered through a 0.22 μm membrane for analysis.

HMOs were separated and quantified using a Shimadzu liquid chromatography-tandem mass spectrometry (LC-MS/MS) system (LCMS-8050,Shimadzu Corporation, Kyoto, Japan). The LC system was equipped with an Alphasil XD-Amide column (4.6 mm × 250 mm, 5 μm, 100 Å, Acchrom, Beijing, China). The injection volume was 5 μL, and the mobile phases consisted of 10 mM ammonium formate in water (A) and acetonitrile (B). A gradient elution was applied at a flow rate of 1 mL/min. Detection was performed using a triple quadrupole mass spectrometer (ESI-QqQ, Shimadzu Corporation, Kyoto, Japan) with electrospray ionization (ESI) in positive ion mode. The MS system operated in multiple reaction monitoring (MRM) mode. The concentration of each HMO in breast milk was determined using an external standard calibration method.

The limit of detection (LOD) and limit of quantification (LOQ) were 0.0025 μg/mL and 0.0083 μg/mL for 2′-FL, 3-FL, LNT, LNnT, LNFP I, and 3′-SL. The LOD and LOQ for LNDFH II were 0.0020 μg/mL and 0.0670 μg/mL, respectively, and 0.0030 μg/mL and 0.0100 μg/mL, respectively, for LSTa. The squares of the correlation coefficients were all over 0.99. The sample recovery rates for low, medium, and high concentrations were in the range of 94.2% to 109.1%, and the relative standard deviations were less than 5%.

### 2.3. Metabolic Factors

In this study, several maternal metabolic factors were assessed during the interview through physical examination, including current weight, height, systolic blood pressure (SBP), diastolic blood pressure (DBP), heart rate (HR), blood glucose (GLU), triglycerides (TG), total cholesterol (TC), high-density lipoprotein cholesterol (HDL-C), and low-density lipoprotein cholesterol (LDL-C). Additionally, maternal pre-pregnancy weight was obtained from the Medical Birth Registry. Current maternal weight and height were measured by healthcare professionals using a calibrated stadiometer and weighing scale. Weight was recorded in kilograms (kg), and height was recorded in centimeters (cm). Maternal pre-pregnancy and current body mass index (BMI) were calculated as weight (kg) divided by the square of height (m^2^). SBP, DBP, and HR were measured using an automated blood pressure monitor, with SBP and DBP expressed in millimeters of mercury (mmHg) and HR in beats per minute (bpm). Fasting blood samples were collected after a 12 h overnight fast. Blood GLU and lipid profiles (TG, TC, HDL-C, and LDL-C) were analyzed using an AU680 Clinical Chemistry Analyzer (Beckman Coulter, Brea, CA, USA) and standardized diagnostic kits (Fosun, Shanghai, China), with concentrations reported in millimoles per liter (mmol/L). All clinical laboratory analyses were performed by the local children’s hospital. Internal quality control was ensured using standardized control materials and the Westgard multi-rule quality control protocol. Furthermore, the hospital’s clinical laboratory routinely participates in external quality assessments administered by the Hunan Provincial Clinical Laboratory Center and has received certification for compliance.

### 2.4. Exposure Variables

Sociodemographic data were collected via questionnaire during the interview. Variables included the following: maternal age (below 25 years, 25 to 29 years, 30 to 34 years, and 35 years and above), lactation period (20 to 60 days, 61 to 90 days, 91 to 150 days, and 151 to 180 days), occupation (including laborers and farmers, office staff, private industry and commerce, and others), education (including junior school and below, high and secondary technical school, and college and above), family annual income (including below 20,000 yuan, 20,000 to 50,000 yuan, and 50,000 yuan and above), and parity (including once, twice, and three times and above).

### 2.5. Statistical Analysis

Data analysis was performed using SPSS 25.0 (IBM Corp., Chicago, IL, USA). Categorical variables are expressed as frequencies and percentages, while continuous variables are summarized as median and interquartile range (IQR) due to their non-normal distribution. The Kruskal–Wallis test was used to assess variations in HMO concentrations across maternal age and lactation period categories. Differences in HMO concentrations and maternal metabolic factors between secretor and non-secretor groups were evaluated using the Mann–Whitney test. Spearman’s rank correlation analysis was employed to examine associations between HMOs and maternal metabolic factors. Linear regression analysis was conducted to further explore the relationship between metabolic factors and HMOs, with results reported as β coefficients and 95% confidence intervals (CIs). Eight individual HMO concentrations were separately included as dependent variables. Each metabolic indicator (pre-pregnancy BMI, current BMI, TC, TG, HDL-C, LDL-C, GLU, SBP, DBP, and HR) was entered as an independent variable in separate models, adjusted for maternal age and lactation period. Additionally, the mediating effects of maternal metabolic factors on HMO concentrations were assessed using the SPSS PROCESS macro, with mediation tested via the nonparametric percentile bootstrap method (5000 iterations). A two-tailed *p*-value < 0.05 was considered statistically significant.

## 3. Results

### 3.1. Participant Demographic Characteristics

A total of 196 mothers from Central South China participated in the study. The majority of participants (72.96%) were aged 25 to 34 years (40.31% aged 25 to 29 years and 32.65% aged 30 to 34 years). Most mothers were in the lactation period of 61 to 150 days (26.53% at 61 to 90 days and 30.10% at 91 to 150 days). Regarding occupational distribution, 37.24% of mothers were office staff. More than half of the participants had attained college-level education or higher, and similarly, over half reported an annual household income of ≥50,000 yuan. Nearly half of the mothers (46.94%) were primiparous (having given birth once). Complete demographic characteristics of the study participants are presented in Table 1.

### 3.2. Oligosaccharides in Breast Milk

A total of eight HMOs were detected, comprising four neutral fucosylated HMOs, 2′-FL, 3-FL, LNFP I, and LNDFH II; two neutral HMOs, LNT and LNnT; and two acidic HMOs, 3′-SL and LSTa. The results revealed that the HMO composition in Central South China was predominantly neutral fucosylated HMOs (Figure 1A). Significant variations were observed in several HMOs based on maternal age (2′-FL, LNT, LNnT, LNFP I, LNDFH II, and LSTa; all *p* < 0.05; Figure 2A,B) and lactation period (3-FL, LNFP I, 3′-SL, and LSTa; all *p* < 0.05; Figure 2C,D). A distinct bimodal distribution of 2′-FL levels was observed, with a clear separation threshold at 150 μg/mL (Appendix A). Based on this threshold, samples with 2′-FL concentrations ≥ 150 μg/mL were classified as secretors (82.05%). Samples below 150 μg/mL were classified as non-secretors (17.95%). Compositional analysis revealed that secretor milk was predominantly composed of 2′-FL (45.86%), while non-secretor milk primarily contained 3-FL (55.40%) (Figure 1A). Comparative analysis showed there were significantly higher concentrations (all *p* < 0.05) of 2′-FL, LNFP I, LNDFH II, and LNnT in secretor versus non-secretor milk, and there were significantly lower concentrations (*p* < 0.05) of 3-FL and LNT (Figure 1B).

### 3.3. Association of Maternal Metabolic Factors with Oligosaccharides

The median (IQR) values for pre-pregnancy BMI and current BMI were 21.56 (19.82–23.92) and 23.11 (21.08–25.71), respectively. Maternal metabolic parameters—including SBP, DBP, HR, GLU, TG, CHOL, HDL-C, and LDL-C—are presented in Table 2. No significant differences in these metabolic factors were observed between secretor and non-secretor mothers. Spearman rank correlation analysis was performed to assess the relationships between maternal metabolic factors and HMOs. Negative correlations were identified between maternal pre-pregnancy BMI, current BMI, TC, TG, and LDL levels and 3′-SL, whereas positive correlations were found between maternal HR and 3-FL, LNDFH II, and 3′-SL (all *p* < 0.05; Figure 3A). However, these associations were only significant in secretor mothers (Figure 3B) and were absent in non-secretor mothers (Figure 3C). Additionally, in secretor mothers, 3-FL was associated with maternal TC, TG, LDL, GLU, and HR (Figure 3B). In contrast, among non-secretor mothers, 2′-FL, LNFP I, and LNDFH II exhibited negative associations with several maternal metabolic factors (Figure 3C).

Table 3 and Table 4 present the associations between maternal metabolic factors and HMOs, analyzed using linear regression after adjusting for maternal age and lactation period. Table 3 reveals that the LNDFH II concentration was positively associated with HR (β = 0.17, 95% CI: 0.03–0.31, *p* = 0.01). Table 4 indicates that the LNnT concentration was positively associated with both SBP (β = 0.16, 95% CI: 0.02–0.30, *p* = 0.03) and DBP (β = 0.19, 95% CI: 0.05–0.33, *p* = 0.01). Additionally, 3′-SL concentration showed a negative association with pre-pregnancy BMI (β = −0.16, 95% CI: −0.29 to −0.03, *p* = 0.02) and current BMI (β = −0.14, 95% CI: −0.27 to −0.01, *p* = 0.04), while exhibiting a positive association with maternal HR (β = 0.14, 95% CI: 0.01–0.27, *p* = 0.04).

In non-secretor mothers, 2′-FL was positively associated with maternal TC, LDL, and DBP. LNFP I was positively associated with maternal SBP and DBP. LNT and LNnT were positively associated with DBP (Appendix A). Among secretor mothers, pre-pregnancy BMI was negatively correlated with 3′-SL, while maternal HR showed positive correlations with 3-FL, LNDFH II, and 3′-SL (Appendix A). Furthermore, we analyzed the associations between metabolic factors, secretor status, and concentrations of LNDFH II and 3′-SL using linear regression, with adjustments for maternal age and lactation period. As shown in Figure 4, maternal secretor status demonstrated a stronger positive association with LNDFH II levels compared to HR. However, no significant associations were observed between secretor status and 3′-SL concentrations.

### 3.4. Mediating Effect of Maternal Metabolic Factors with Oligosaccharides

In this study, Spearman rank correlation analysis revealed that 3′-SL exhibited negative associations with several metabolic factors, including pre-pregnancy BMI, current BMI, TC, TG, and LDL. However, after adjusting for maternal age and lactation period in linear regression analysis, 3′-SL remained negatively associated only with pre-pregnancy BMI and current BMI. To explore whether TC, TG, and LDL mediated the relationship between maternal BMI and 3′-SL, we conducted mediation analysis (Figure 5). The results indicated that TG and LDL mediated the association between pre-pregnancy BMI and 3′-SL. LDL mediated the association between current BMI and 3′-SL. Additionally, TC mediated the associations of maternal age and lactation period with 3′-SL (Appendix A).

## 4. Discussion

In this cross-sectional study, we analyzed the concentrations of eight major HMOs in 196 breast milk samples collected from South Central China. Our results demonstrated that the HMO profile in this region was predominantly composed of neutral fucosylated HMOs. Significant variations in the levels of several HMOs were observed across different maternal age groups and lactation stages. Notably, 3′-SL exhibited distinct associations with multiple maternal metabolic factors, with differential patterns between secretor and non-secretor mothers. Specifically, in secretor mothers, 3′-SL was significantly associated with pre-pregnancy BMI and HR. Mediation analysis further revealed that TG and LDL mediated the relationship between maternal pre-pregnancy BMI and 3′-SL, while TC mediated the associations of maternal age and lactation period with 3′-SL.

Our findings complement existing data on HMO composition in breast milk from South Central China. This study revealed that breast milk from this region primarily consists of neutral fucosylated HMOs, with approximately 20% of mothers classified as non-secretors. Among secretor mothers, 2′-FL was the dominant HMO, whereas 3-FL predominated in non-secretor mothers. These results align with previously reported HMO profiles in other Chinese populations [20,21]. We further characterized the variation patterns of HMOs across maternal age and lactation stages. Consistent with prior research, 3-FL levels increased during lactation [20]. However, 2′-FL exhibited a distinct dynamic: its concentration declined from 20 to 150 days postpartum, followed by an increase between 151 and 180 days. Many studies have reported a decreasing trend for 2′-FL during lactation [11,22]. Additionally, most HMOs displayed a declining trend with advancing maternal age, though existing studies report inconsistent conclusions due to confounding factors such as maternal BMI, lactation stage, and parity [23,24].

Regarding maternal metabolic factors, our findings differ from previous studies. While some research has reported associations between maternal status and specific HMOs (e.g., 2′-FL, LNFP III, LNnT, fucosylated HMOs, and total HMOs) [13,24,25], these studies presented inconsistent results. In contrast, we observed that maternal BMI was only associated with 3′-SL, with no significant links to other HMOs. For 3′-SL, prior findings have also been conflicting. One study reported higher 3′-SL levels in overweight mothers compared to obese mothers [26], aligning with our observation of an inverse relationship between 3′-SL and maternal BMI. Our earlier work demonstrated that maternal dietary patterns influence both maternal weight [27] and breast milk macronutrient composition [28]. Other studies suggest that maternal diet directly affects HMO profiles [29], leading us to hypothesize that the BMI–HMO association may be mediated by dietary intake. Notably, the correlation between 3′-SL and maternal BMI was exclusive to secretor mothers. Another study found that overweight non-secretors had significantly lower 3′-SL levels than obese non-secretors [30]. These founding suggest the role of maternal genetic factors (e.g., secretor status) in modulating this relationship. Furthermore, some research supports the hypothesis that HMO composition arises from interactions between maternal genetics and diet [31].

Oligosaccharides are not exclusive to human milk. Previous studies have demonstrated that HMOs are present in the maternal systemic circulation during pregnancy and are linked to cardiovascular metabolic factors [32,33,34]. Our study expands upon these findings by examining the influence of maternal cardiovascular metabolic factors on HMO composition. We found that NDFH II and 3′-SL concentration were positively associated with maternal HR—a novel finding not previously reported. While limited data exist on the direct relationship between HMOs and HR, we speculate that this association may be related to gut microbiota metabolism. Many studies have confirmed that the gut microbiota influence HR, and oligosaccharides, as prebiotics, may improve the microbial balance [35,36]. Additionally, the observed link was exclusive to secretor mothers. For LNDFH, maternal secretor status showed a stronger positive association with LNDFH II levels than HR. These findings suggested a potential role for maternal genetic factors. However, no significant associations were observed between secretor status and 3′-SL concentrations by linear regression analysis. The possible reasons may involve the relatively small sample size of non-secretor mothers in this study, or potential interaction effects between maternal secretor status and metabolic factors on 3′-SL concentrations. We also identified associations between blood pressure and several HMOs, possibly mediated by intestinal flora. Prior research supports this mechanism, indicating that oligosaccharides may help regulate blood pressure by modulating gut microbiota [37]. Contrary to other studies reporting connections between maternal glucose metabolism and HMOs [32,38], we found no association between maternal glucose (GLU) levels and HMOs—likely because our study excluded participants with diabetes.

Additionally, our study is among the first to explore the association between circulating lipids and HMOs. Our results indicated that maternal TC, TG, and LDL showed negative correlations with 3′-SL in unadjusted analyses. A previous study displayed that maternal serum 3′-SL and 2′-FL were associated with lipid profiles during gestation [16]. Williams et al. also reported associations between breast milk fatty acids and HMO compositions [39]. This mechanism might involve gut microbiota-mediated metabolism, whereby oligosaccharides regulate bacterial proliferation and subsequently alter lipid metabolism pathways [40]. However, after adjusting for maternal age and lactation period, these associations lost statistical significance, highlighting the confounding influence of these factors. Moreover, the mediation analysis revealed that TC acts as a mediating factor in the association of lactation period with 3′-SL, but they masked the direct effect of lactation period on 3′-SL. These results also provide additional information that the reasons for the impact of lactation period on HMOs may involve lipid metabolism. Furthermore, we found that TG and LDL mediated the associations of maternal BMI with 3′-SL, especially for pre-pregnancy BMI. Several animal studies found an effect of 2′-FL on weight gain by improving lipid metabolism, which also involved changes in the gut microbiota and metabolites [41,42]. However, no prior studies have directly examined HMOs in breast milk and maternal serum lipids, limiting mechanistic conclusions. Thus, we cautiously propose that maternal BMI’s influence on 3′-SL may involve lipid metabolism, though further research is needed.

Our study provides the first comprehensive data on HMO concentrations in lactating women from South Central China, along with novel insights into how maternal metabolic factors may influence HMO profiles. These findings could inform future guidelines for personalized maternal and infant nutrition. However, this study has several limitations. As maternal diet is a known confounding factor affecting weight status, metabolic parameters, and HMO composition [24,43], its absence in our analysis may have influenced the results. Furthermore, this study did not assess infant growth and development, leaving open questions about how maternal metabolic–HMO associations might impact infant health—a critical area for future research.

## 5. Conclusions

The variations of several HMOs among mothers from Central South China were asso-ciated with maternal age and lactation period. The concentration of 3′-SL was negatively correlated with maternal pre-pregnancy BMI. The potential mechanism underlying the influence of maternal BMI on 3′-SL levels may involve maternal lipid metabolism and genetic factors. However, these findings were based on a cross-sectional design, and the causal relationships need to be further validated by prospective studies.

## Figures and Tables

**Figure 1 nutrients-17-01480-f001:**
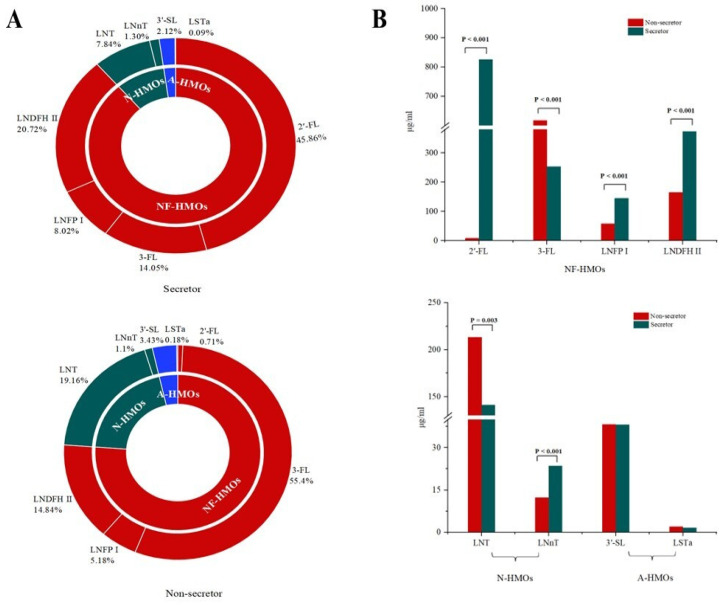
Composition and comparative analysis of HMOs. (**A**) Proportional distribution of HMO subtypes. (**B**) Concentration profiles of eight individual HMOs stratified by maternal secretor status. Abbreviations: NF-HMOs, neutral fucosylated HMOs; *N*-HMOs, neutral non-fucosylated HMOs; A-HMOs, acidic HMOs. Statistical significance was determined with Mann–Whitney U tests (*p* < 0.05).

**Figure 2 nutrients-17-01480-f002:**
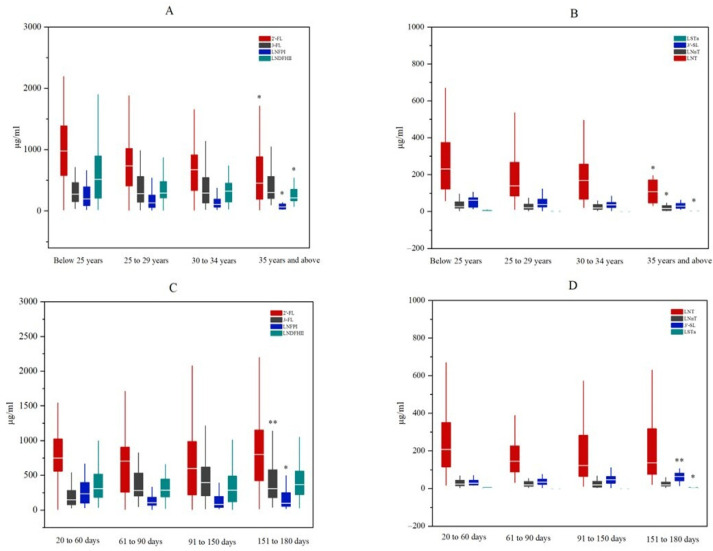
Concentration profiles of HMOs stratified by maternal age and lactation period. (**A**) Neutral fucosylated HMOs across maternal age groups; (**B**) Neutral and acidic HMOs across maternal age groups; (**C**) Neutral fucosylated HMOs across lactation periods; (**D**) Neutral and acidic HMOs across lactation periods. Statistical significance: * *p* < 0.05, ** *p* < 0.001 (Kruskal–Wallis test).

**Figure 3 nutrients-17-01480-f003:**
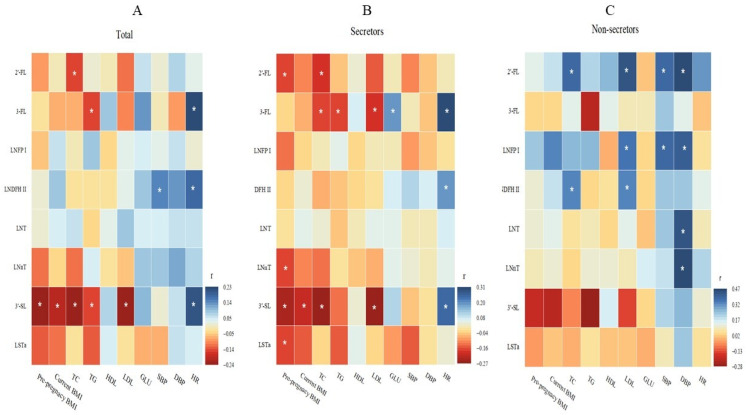
The correlation of HMOs with maternal metabolic factors. (**A**) All mothers; (**B**) Secretors; (**C**) Non-secretors; r: correlation coefficient; *: *p* < 0.05 by Spearman rank correlation analysis.

**Figure 4 nutrients-17-01480-f004:**
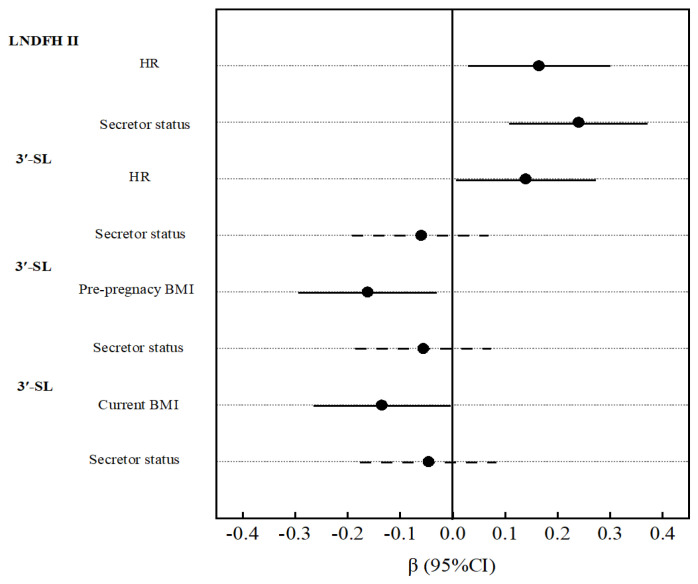
Associations of metabolic factors and secretor status with LNDFH II and 3′-SL concentrations in the total study population, based on linear regression analysis. LNDFH II and 3′-SL concentrations were separately included as dependent variables. Metabolic indicators and secretor status were entered as independent variables in separate models, adjusted for maternal age and lactation period. The solid line indicates statistical significance (*p* < 0.05).

**Figure 5 nutrients-17-01480-f005:**
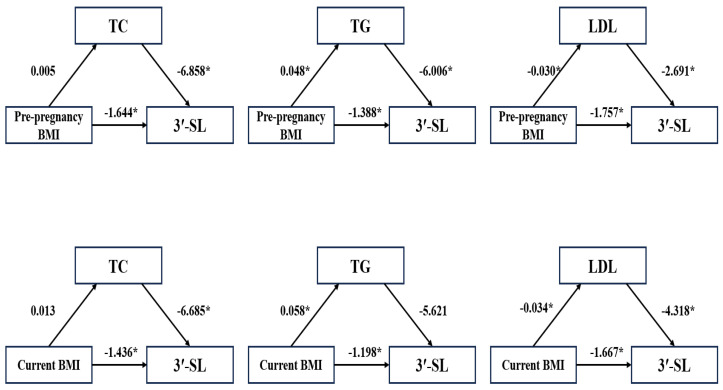
The mediating effect analysis of TC, TG, and LDL on the association of maternal BMI with 3′-SL. HR as a covariate was included in the mediating effect model. The number above the arrow represents the unstandardized regression coefficient; *: *p* < 0.05 for correlation coefficient.

**Table 1 nutrients-17-01480-t001:** Characteristics of the participating mothers from Central South China.

Characteristics		N (%)
Maternal age		
	Below 25 years	31 (15.82)
	25 to 29 years	79 (40.31)
	30 to 34 years	64 (32.65)
	35 years and above	22 (11.22)
Lactating period		
	20 to 60 days	48 (24.49)
	61 to 90 days	52 (26.53)
	91 to 150 days	59 (30.10)
	151 to 180 days	37 (18.88)
Occupation		
	Laborers and farmers	20 (10.20)
	Office staff	73 (37.24)
	Private industry and commerce	34 (17.35)
	Other	69 (35.20)
Education		
	College and above	102 (52.04)
	High and secondary technical school	66 (33.67)
	Junior school and below	28 (14.29)
Family income		
	50,000 yuan and above	109 (55.61)
	20,000 to 50,000 yuan	53 (27.04)
	Below 20,000 yuan	34 (17.35)
Parity		
	Once	92 (46.94)
	Twice	81 (41.33)
	Three times and above	23 (11.73)

**Table 2 nutrients-17-01480-t002:** Characteristics of maternal metabolic factors in Central South China (median [IQR]).

Metabolic Factors	Total		Non-Secretor		Secretor		*p*
Pre-pregnancy BMI	21.56	(19.82, 23.92)	21.48	(19.56, 23.73)	21.64	(19.84, 23.93)	0.774
Current BMI	23.11	(21.08, 25.71)	22.77	(20.31, 24.16)	23.36	(21.16, 25.90)	0.219
TC mmol/L	4.98	(4.25, 5.43)	5.03	(4.20, 5.88)	4.98	(4.26, 5.34)	0.291
TG mmol/L	0.92	(0.70, 1.43)	0.89	(0.65, 1.55)	0.92	(0.75, 1.41)	0.542
HDL mmol/L	1.59	(1.41, 1.79)	1.61	(1.43, 1.77)	1.59	(1.40, 1.80)	0.473
LDL mmol/L	2.93	(2.50, 3.44)	2.94	(2.47, 3.59)	2.93	(2.50, 3.42)	0.449
GLU mmol/L	4.77	(4.44, 5.22)	4.72	(4.44, 5.10)	4.81	(4.44, 5.23)	0.426
SBP mmHg	114	(103, 122)	111	(99, 122)	115	(105, 123)	0.197
DBP mmHg	71	(64, 79)	68	(61, 76)	71	(64, 80)	0.080
HR bmp	80	(71, 88)	80	(72, 87)	80	(71, 89)	0.899

*p* from Mann–Whitney test.

**Table 3 nutrients-17-01480-t003:** The association of maternal metabolic factors with neutral fucosylated HMOs among total participant mothers by linear regression analysis.

Metabolic Factors	2′-FL			3-FL			LNFP I			LNDFH II		
	β (95% CI)		*p*	β (95% CI)		*p*	β (95% CI)		*p*	β (95% CI)		*p*
Pre-pregnancy BMI	−0.07	(−0.22, 0.07)	0.29	0.01	(−0.13, 0.15)	0.86	−0.09	(−0.23, 0.05)	0.20	−0.02	(−0.16, 0.12)	0.73
Current BMI	−0.02	(−0.16, 0.12)	0.77	−0.05	(−0.19, 0.09)	0.52	−0.03	(−0.17, 0.11)	0.64	0.05	(−0.09, 0.19)	0.45
TC	−0.12	(−0.27, 0.02)	0.10	0.05	(−0.10, 0.20)	0.48	0.02	(−0.13, 0.17)	0.78	0.10	(−0.04, 0.25)	0.16
TG	0.00	(−0.14, 0.14)	1.00	−0.11	(−0.25, 0.03)	0.13	−0.02	(−0.16, 0.12)	0.74	−0.02	(−0.16, 0.12)	0.77
HDL	−0.07	(−0.20, 0.07)	0.35	0.13	(−0.01, 0.27)	0.06	−0.03	(−0.17, 0.11)	0.67	−0.01	(−0.15, 0.12)	0.84
LDL	−0.11	(−0.26, 0.04)	0.15	0.02	(−0.13, 0.16)	0.84	0.02	(−0.12, 0.17)	0.76	0.14	(−0.01, 0.28)	0.07
GLU	0.01	(−0.13, 0.15)	0.88	0.04	(−0.10, 0.18)	0.61	0.04	(−0.10, 0.18)	0.58	0.11	(−0.03, 0.24)	0.13
SBP	0.03	(−0.11, 0.17)	0.72	0.01	(−0.13, 0.15)	0.93	0.03	(−0.11, 0.17)	0.68	0.13	(0.00, 0.27)	0.06
DBP	0.10	(−0.04, 0.23)	0.18	−0.06	(−0.20, 0.08)	0.40	0.08	(−0.06, 0.22)	0.25	0.13	(−0.01, 0.26)	0.07
HR	0.04	(−0.10, 0.18)	0.61	0.09	(−0.05, 0.23)	0.22	0.00	(−0.14, 0.14)	0.98	0.17	(0.03, 0.31)	0.01

The model adjusted for maternal age and lactation period; TC, TG, HDL, LDL and GLU expressed by mmol/L; SBP and DBP expressed by mmHg; HR expressed by bmp; 2′-FL, 3-FL, LNFP I, and LNDFH II expressed by μg/mL.

**Table 4 nutrients-17-01480-t004:** The association of metabolic factors with neutral and acidic HMOs among total participant mothers by linear regression analysis.

Metabolic Factors	LNT			LNnT			3′-SL			LSTa		
	β (95% CI)		*p*	β (95% CI)		*p*	β (95% CI)		*p*	β (95% CI)		*p*
Pre-pregnancy BMI	0.02	(−0.12, 0.17)	0.75	0.02	(−0.13, 0.16)	0.83	−0.16	(−0.29, −0.03)	0.02	−0.05	(−0.19, 0.09)	0.50
Current BMI	0.03	(−0.11, 0.18)	0.66	0.04	(−0.10, 0.18)	0.56	−0.14	(−0.27, −0.01)	0.04	−0.02	(−0.16, 0.12)	0.80
TC	0.05	(−0.10, 0.20)	0.49	−0.06	(−0.21, 0.09)	0.42	−0.11	(−0.24, 0.03)	0.14	−0.02	(−0.17, 0.13)	0.83
TG	0.02	(−0.12, 0.17)	0.75	0.02	(−0.13, 0.16)	0.82	−0.09	(−0.22, 0.04)	0.19	0.03	(−0.11, 0.18)	0.63
HDL	0.04	(−0.10, 0.18)	0.57	−0.04	(−0.18, 0.10)	0.54	0.04	(−0.09, 0.17)	0.59	0.02	(−0.12, 0.16)	0.75
LDL	0.06	(−0.09, 0.21)	0.46	−0.04	(−0.19, 0.11)	0.59	−0.11	(−0.25, 0.03)	0.11	−0.03	(−0.18, 0.12)	0.73
GLU	0.03	(−0.11, 0.18)	0.64	0.09	(−0.05, 0.23)	0.22	0.07	(−0.06, 0.20)	0.27	0.05	(−0.09, 0.19)	0.48
SBP	0.10	(−0.04, 0.25)	0.15	0.16	(0.02, 0.30)	0.03	0.01	(−0.12, 0.14)	0.88	0.05	(−0.09, 0.19)	0.47
DBP	0.13	(−0.01, 0.27)	0.07	0.19	(0.05, 0.33)	0.01	0.07	(−0.07, 0.20)	0.32	0.11	(−0.04, 0.25)	0.14
HR	0.01	(−0.14, 0.15)	0.93	−0.04	(−0.19, 0.10)	0.57	0.14	(0.01, 0.27)	0.04	−0.02	(−0.16, 0.13)	0.81

The model adjusted for maternal age and lactation period; TC, TG, HDL, LDL, and GLU expressed by mmol/L; SBP and DBP expressed by mmHg; HR expressed by bmp; LNT, LNnT, 3′-SL and LSTa expressed by μg/mL.

## Data Availability

The original contributions presented in this study are included in the article/Appendix A. Further inquiries can be directed to the corresponding author.

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
