# Peer review of "Impact of Maternal Metabolic Status on Human Milk Oligosaccharide Composition: A Population-Based Cross-Sectional Study in Central South China"

_nutrients, 2025, doi:10.3390/nu17091480_

Round 1
Reviewer 1 Report
Comments and Suggestions for Authors
The manuscript "Impact of Maternal Metabolic Status on Human Milk Oligosaccharide Composition: A Population-Based Cross-sectional Study in Central South China" provides a lot of information on the oligosaccharide composition of human milk (HMOs), reporting data on eight of them, which would represent the quantitatively most prevalent and significant proportion. The manuscript is clear in its various parts, but the figures are very difficult to decipher because they are too small and have microscopic print. Table 3 has many discrepancies and errors from what is written in the text. In addition, both Tables 3 and 4 should give the units of measurement for each parameter. It is not clear why Table 4 is described in the text before Table 3.
Please enter the units of the variables in Table 3 and 4.
How do the authors comment that higher HR is directly associated with both LNDFH II and 3′-SL?
In the model adjusted for maternal age and lactation period, TC, TG and LDL are not significantly associated with 3′-SL (see Table 4). Therefore, theoretically, there would be no justification to perform a mediation analysis with these variables. In the mediation analysis, the direct effect of TG and LDL becomes significant, although it was not significant in the statistical model shown in Table 4. I would like to ask the authors to comment on this point.
The mediation analysis should also be corrected for HR, as it is the only other variable significantly associated with 3′-SL.
With regard to the conclusions, I believe that the qualitative and quantitative differences reported for HMOs during pregnancy should be validated by a prospective study evaluating the health of the children of these women over time, and it is not definitive what can be observed in a cross-sectional study.
In general, when commenting on the study, one can only take note of the data provided on the composition and concentration of oligosaccharides, but also of the fact that, as the authors correctly report, the literature on this subject is very mixed. It is possible that a positive bias towards breast milk, which would always have a beneficial effect on the health of both the fetus and the mother, plays a role in this area of research. In fact, looking at the data from the study, only the BMI in pregnancy and at the time of detection has an inverse correlation with 3'SL, thus a trend that can be considered positive for health. The other parameters with a significant positive correlation with the HMOs tested, which increased, cannot be considered as indices of good maternal health (heart rate, systolic and diastolic blood pressure, total cholesterol, LDL cholesterol, triglycerides). However, evaluation in pregnancy is difficult because what may be detrimental to maternal health may be functional for optimal fetal growth. In this sense, HMOs may reflect a natural tendency to preserve the health of the offspring, even at the expense of the mother's health. An extreme example of a similar process might be the maintenance of the macronutrient and calcium composition in the milk of severely malnourished mothers. Leaving aside this "philosophical" digression, it would be interesting if studies on the micronutrient composition of breast milk also assessed the growth and health status of the infant.
Author Response
Dear Reviewer
We are very grateful for your advices and comments about our manuscript. Those were helpful to improve our paper. We had made a point-by-point revision and hope to meet with approval. Revised portion are marked in red in the paper.The responds to the reviewer’s comments are as following:
The manuscript "Impact of Maternal Metabolic Status on Human Milk Oligosaccharide Composition: A Population-Based Cross-sectional Study in Central South China" provides a lot of information on the oligosaccharide composition of human milk (HMOs), reporting data on eight of them, which would represent the quantitatively most prevalent and significant proportion. The manuscript is clear in its various parts, but the figures are very difficult to decipher because they are too small and have microscopic print. Table 3 has many discrepancies and errors from what is written in the text. In addition, both Tables 3 and 4 should give the units of measurement for each parameter. It is not clear why Table 4 is described in the text before Table 3.
Response: Thank you for the reviewers' suggestions. Based on the reviewers' comments, we have further adjusted the sizes of Figures 1, 2 and 3 in the article, as shown in Figures 1-3. Additionally, we have revised the descriptions of Tables 3 and 4.
The revised descriptions are as follows:
Table 3 reveals that LNDFH II concentration was positively associated with HR (β = 0.17, 95% CI: 0.03–0.31, P = 0.01). Table 4 indicates that LNnT concentration was positively associated with both SBP (β = 0.16, 95% CI: 0.02–0.30, P = 0.03) and DBP (β = 0.19, 95% CI: 0.05–0.33, P = 0.01). Additionally, 3′-SL concentration showed a negative association with pre-pregnancy BMI (β = -0.16, 95% CI: -0.29 to -0.03, P = 0.02) and current BMI (β = -0.14, 95% CI: -0.27 to -0.01, P = 0.04), while exhibiting a positive association with maternal HR (β = 0.14, 95% CI: 0.01–0.27, P = 0.04).
Please enter the units of the variables in Table 3 and 4.
Response: According to the reviewers' suggestions, we have added the units of the variables in the notes of Tables 3 and 4.
The descriptions in the notes are as follows:
Table 3: TC, TG, HDL, LDL and GLU expressed by mmol/L; SBP and DBP expressed by mmHg; HR expressed by bmp; 2′-FL, 3-FL, LNFP I, and LNDFH II expressed by μg/mL
Table 4: TC, TG, HDL, LDL, and GLU expressed by mmol/L; SBP and DBP expressed by mmHg; HR expressed by bmp; LNT, LNnT, 3′-SL and LSTa expressed by μg/mL
How do the authors comment that higher HR is directly associated with both LNDFH II and 3′-SL?
Response:Thank you for the reviewer's suggestion. We have added comments on the association of LNDFH II and 3′-SL with HR in the discussion section.
The comments are as follows:
We found that NDFH II and 3′-SL concentration were positively associated with maternal HR—a novel finding not previously reported. While limited data exist on the direct relationship between HMOs and HR, we speculate that this association may be related to gut microbiota metabolism. Many studies have confirmed that gut microbiota influence HR, and oligosaccharides, as prebiotics, may improve microbial balance [35,36]. Additionally, the observed link was exclusive to secretor mothers, suggesting a potential role for maternal genetic factors. We also identified associations between blood pressure and several HMOs, possibly mediated by intestinal flora. Prior research supports this mechanism, indicating that oligosaccharides may help regulate blood pressure by modulating gut microbiota [37].
In the model adjusted for maternal age and lactation period, TC, TG and LDL are not significantly associated with 3′-SL (see Table 4). Therefore, theoretically, there would be no justification to perform a mediation analysis with these variables. In the mediation analysis, the direct effect of TG and LDL becomes significant, although it was not significant in the statistical model shown in Table 4. I would like to ask the authors to comment on this point.
Response:We sincerely appreciate the reviewer's suggestion. In response, we have added comments regarding the association between TC, TG, LDL and 3′-SL in the Discussion section.
The added comments as follows:
Additionally, our study is among the first to explore the association between cir-culating lipids and HMOs. Our results indicated that maternal TC, TG, and LDL showed negative correlations with 3′-SL in unadjusted analyses. A previous study displayed that maternal serum 3′-SL and 2′-FL were associated with lipid profiles in gestation [16]. Williams et al. also reported associations between breast milk fatty acids and HMO compositions [39]. This mechanism might involve gut microbiota-mediated metabolism, whereby oligosaccharides regulate bacterial proliferation and subsequently alter lipid metabolism pathways[40]. However, after adjusting for maternal age and lactation period, these associations lost statistical significance, highlighting the confounding influence of these factors. Moreover, the mediation analysis revealed that TC act as a mediating factor in the association of lactation period with 3′-SL, but they masked the direct effect of lactation period on 3′-SL. These results also provide additional information that the reasons for the impact of lactation period on HMOs may involve lipid metabolism. Furhtermore, we found that TG and LDL mediated the associations of maternal BMI with 3′-SL, especially for pre-pregnacy BMI. Several animal studies present an effect of 2′-FL on weight gain by improving lipid metabolism , which also involved the changes of the gut microbiota and metabolites [41,42]. However, no prior studies have directly examined HMOs in breast milk and maternal serum lipids, limiting mechanistic conclusions. Thus, we cautiously propose that maternal BMI’s influence on 3′-SL may involve lipid metabolism, though further research is needed.
The mediation analysis should also be corrected for HR, as it is the only other variable significantly associated with 3′-SL.
Response: Thank you for the reviewers' suggestions. In the mediation effect model, we further included heart rate as a covariate. The results of the mediation effect after the revision are shown in Figure 4 and Supplementary Figure S2.
The results indicated that TG and LDL mediated the association between pre-pregnancy BMI and 3′-SL. LDL mediated the association between current BMI and 3′-SL(Figure 4). Additionally, TC mediated the associations of maternal age and lactation period with 3′-SL (Supplementary Figure S2).
With regard to the conclusions, I believe that the qualitative and quantitative differences reported for HMOs during pregnancy should be validated by a prospective study evaluating the health of the children of these women over time, and it is not definitive what can be observed in a cross-sectional study.
Response:We thank the reviewers for their valuable suggestions. Based on these advices, we have added a description of limitations in the Conclusion section
The description as follows:
However, these findings were depended on a cross-sectional design, and the causal relationships should be further confirmed by prospective studies.
In general, when commenting on the study, one can only take note of the data provided on the composition and concentration of oligosaccharides, but also of the fact that, as the authors correctly report, the literature on this subject is very mixed. It is possible that a positive bias towards breast milk, which would always have a beneficial effect on the health of both the fetus and the mother, plays a role in this area of research. In fact, looking at the data from the study, only the BMI in pregnancy and at the time of detection has an inverse correlation with 3'SL, thus a trend that can be considered positive for health. The other parameters with a significant positive correlation with the HMOs tested, which increased, cannot be considered as indices of good maternal health (heart rate, systolic and diastolic blood pressure, total cholesterol, LDL cholesterol, triglycerides). However, evaluation in pregnancy is difficult because what may be detrimental to maternal health may be functional for optimal fetal growth. In this sense, HMOs may reflect a natural tendency to preserve the health of the offspring, even at the expense of the mother's health. An extreme example of a similar process might be the maintenance of the macronutrient and calcium composition in the milk of severely malnourished mothers. Leaving aside this "philosophical" digression, it would be interesting if studies on the micronutrient composition of breast milk also assessed the growth and health status of the infant.
Response: We sincerely appreciate the reviewer's insightful comments regarding the limitations of our study and suggestions for future research directions. Maternal nutritional status during pregnancy and lactation and its impact on infant growth and development have always been a key focus of our research group. However, while our previous studies primarily examined the effects of maternal nutritional status on breast milk composition, we have yet to thoroughly investigate its implications for infant growth and development. The reviewer's comments have provided us with valuable guidance for future studies, and we are truly grateful for these constructive suggestions.
We tried our best to improve the manuscript and made some changes in the manuscript. We appreciate for Editors and Reviewers’ warm work earnestly, and hope that the correction will meet with approval. Once again, thank you very much for your comments and suggestions.
Yours
Sincerely
Zhi Huang
Reviewer 2 Report
Comments and Suggestions for Authors
This manuscript presents a well-conducted population-based cross-sectional study exploring the association between maternal metabolic factors and the composition of human milk oligosaccharides (HMOs) in Central South China. The topic is timely and relevant, considering the increasing interest in optimizing early-life nutrition and the potential role of HMOs in infant health.
- Introduction Clarity: The introduction would benefit from improved language clarity and grammar. For example, the sentence "Breast milk as the most natural and nutritious food source for infants was recommended exclusively breastfeeding..." should be revised for better readability.
- Novelty and Literature Comparison: The authors mention discrepancies in prior studies regarding maternal BMI and HMO profiles, but the discussion could benefit from a deeper exploration of why their results may differ and how regional dietary or genetic variations might contribute.
- Statistical Justification: While the use of non-parametric tests is appropriate due to data distribution, the manuscript could include a justification for selecting specific HMOs and whether any correction for multiple testing was applied.
- Discussion Expansion: The findings regarding 3′-SL and its mediation through TG and LDL are interesting. However, more biological interpretation and reference to existing mechanisms or hypotheses in the literature would enhance the impact of the discussion.
- Formatting and Language: The manuscript contains several typographical and formatting errors (e.g., missing spaces, inconsistent citation formatting).
Author Response
Dear Reviewer
We are very grateful for your advices and comments about our manuscript. Those were helpful to improve our paper. We had made a point-by-point revision and hope to meet with approval. Revised portion are marked in red in the paper.The responds to the reviewer’s comments are as following:
This manuscript presents a well-conducted population-based cross-sectional study exploring the association between maternal metabolic factors and the composition of human milk oligosaccharides (HMOs) in Central South China. The topic is timely and relevant, considering the increasing interest in optimizing early-life nutrition and the potential role of HMOs in infant health.
Introduction Clarity: The introduction would benefit from improved language clarity and grammar. For example, the sentence "Breast milk as the most natural and nutritious food source for infants was recommended exclusively breastfeeding..." should be revised for better readability.
Response: We sincerely apologize for the inappropriate language expressions in our manuscript. According to the reviewers' comments, we have further revised the language in our draft to improve its readability
Novelty and Literature Comparison: The authors mention discrepancies in prior studies regarding maternal BMI and HMO profiles, but the discussion could benefit from a deeper exploration of why their results may differ and how regional dietary or genetic variations might contribute.
Response: Thank you for these advices. Based on these suggestions, we have added commentary on the influence of dietary and genetic factors on the association between maternal BMI and the composition of HMOs.
The commentary is as follows:
In contrast, we observed that maternal BMI was only associated with 3′-SL, with no significant links to other HMOs. For 3′-SL, prior findings have also been conflicting. One study reported higher 3′-SL levels in overweight mothers compared to obese mothers [26], aligning with our observation of an inverse relationship between 3′-SL and maternal BMI. Our earlier work demonstrated that maternal dietary patterns in-fluence both maternal weight [27] and breast milk macronutrient composition [28]. Other studies suggest that maternal diet directly affects HMO profiles [29], leading us to hypothesize that the BMI-HMO association may be mediated by dietary intake. Notably, the correlation between 3′-SL and maternal BMI was exclusive to secretor mothers. Another study in which overweight non-secretors had significantly lower 3′-SL levels than obese non-secretors [30]. These founding suggest the role of maternal genetic factors (e.g., secretor status) in modulating this relationship. Furthermore, some researches support that HMO composition arises from interactions between maternal genetics and diet [31].
Statistical Justification: While the use of non-parametric tests is appropriate due to data distribution, the manuscript could include a justification for selecting specific HMOs and whether any correction for multiple testing was applied.
Response: Thank you for these advices. We have revised the description of multivariate regression analysis in the Statistical Methods section based on the authors' recommendations.
Linear regression analysis was conducted to further explore the relationship between metabolic factors and HMOs, with results reported as β coefficients and 95% confi-dence intervals (CIs). Eight individual HMO concentrations were separately included as dependent variables. Each metabolic indicator (pre-pregnancy BMI, current BMI, TC, TG, HDL-C, LDL-C, GLU, SBP, DBP, and HR) was entered as an independent variable in separate models, adjusted for maternal age and lactation period.
The independent variable was entered individually into regression models. Therefore, multiple testing was not conducted in this section.
Discussion Expansion: The findings regarding 3′-SL and its mediation through TG and LDL are interesting. However, more biological interpretation and reference to existing mechanisms or hypotheses in the literature would enhance the impact of the discussion.
Response: According to the reviewers' comments, we have enhanced the mechanistic discussion regarding the TG/LDL and 3′-SL relationship.
The modifications as detailed below:
Additionally, our study is among the first to explore the association between cir-culating lipids and HMOs. Our results indicated that maternal TC, TG, and LDL showed negative correlations with 3′-SL in unadjusted analyses. A previous study dis-played that maternal serum 3′-SL and 2′-FL were associated with lipid profiles in ges-tation [16]. Williams et al. also reported associations between breast milk fatty acids and HMO compositions [39]. This mechanism might involve gut microbiota-mediated metabolism, whereby oligosaccharides regulate bacterial proliferation and subse-quently alter lipid metabolism pathways[40].However, after adjusting for maternal age and lactation period, these associations lost statistical significance, highlighting the confounding influence of these factors. Moreover, the mediation analysis revealed that TC act as a mediating factor in the association of lactation period with 3′-SL, but they masked the direct effect of lactation period on 3′-SL. These results also provide addi-tional information that the reasons for the impact of lactation period on HMOs may involve lipid metabolism. Furhtermore, we found that TG and LDL mediated the asso-ciations of maternal BMI with 3′-SL, especially for pre-pregnacy BMI. Several animal studies present an effect of 2′-FL on weight gain by improving lipid metabolism , which also involved the changes of the gut microbiota and metabolites [41,42]. However, no prior studies have directly examined HMOs in breast milk and maternal serum lipids, limiting mechanistic conclusions. Thus, we cautiously propose that maternal BMI’s in-fluence on 3′-SL may involve lipid metabolism, though further research is needed.
Formatting and Language: The manuscript contains several typographical and formatting errors (e.g., missing spaces, inconsistent citation formatting).
Response: We sincerely apologize for any inadequacies in our manuscript. In accordance with the reviewers' suggestions, we have thoroughly revised both the language and formatting of the manuscript.
We tried our best to improve the manuscript and made some changes in the manuscript. We appreciate for Editors and Reviewers’ warm work earnestly, and hope that the correction will meet with approval. Once again, thank you very much for your comments and suggestions.
Yours
Sincerely
Zhi Huang
Reviewer 3 Report
Comments and Suggestions for Authors
Huang et al. measured several oligosaccharides concentrations in 195 human milk samples and examined their statistical associations with physiological and metabolic factors of mother.
It is known in the previous studies that oligosaccharides concentrations vary with genetic (secretor and non-secretor) and environmental factors of lactating mothers and the present results were consistent with the literature knowledge.
In fact, significant associations were sporadically found in the regression analyses, involving both genetic and non-genetic factors, in this study, which could not be systematically interpreted by the authors: this is because the study was not designed as such. In this sense, the present study is not novel nor original and it just added another similar result on the body of knowledge on variation of oligosaccharide concentrations in human milk.
The followings should be reconsidered.
- There are minor linguistic errors throughout the manuscript. The manuscript must be edited by a professional editor before submission.
- Please consider inclusion of secretor/non-secretor as an independent variable in the multiple regression analyses (Tabs 3 & 4): that would make it possible to explicitly compare the contribution of genetic and metabolic factors to the variation of individual oligosaccharide levels. In other words, the present multiple regressions are not very informative without inclusion of the genetic factor.
- Please specify what are the numbers attached to the arrows in Fig. 4.
Author Response
Dear Reviewer
We are very grateful for your advices and comments about our manuscript. Those were helpful to improve our paper. We had made a point-by-point revision and hope to meet with approval. Revised portion are marked in red in the paper.The responds to the reviewer’s comments are as following:
Huang et al. measured several oligosaccharides concentrations in 195 human milk samples and examined their statistical associations with physiological and metabolic factors of mother.
It is known in the previous studies that oligosaccharides concentrations vary with genetic (secretor and non-secretor) and environmental factors of lactating mothers and the present results were consistent with the literature knowledge.
In fact, significant associations were sporadically found in the regression analyses, involving both genetic and non-genetic factors, in this study, which could not be systematically interpreted by the authors: this is because the study was not designed as such. In this sense, the present study is not novel nor original and it just added another similar result on the body of knowledge on variation of oligosaccharide concentrations in human milk.
The followings should be reconsidered.
- There are minor linguistic errors throughout the manuscript. The manuscript must be edited by a professional editor before submission.
Response: We sincerely apologize for any inadequacies in our manuscript. In accordance with the reviewers' suggestions, we have thoroughly revised both the language and formatting of the manuscript.
- Please consider inclusion of secretor/non-secretor as an independent variable in the multiple regression analyses (Tabs 3 & 4): that would make it possible to explicitly compare the contribution of genetic and metabolic factors to the variation of individual oligosaccharide levels. In other words, the present multiple regressions are not very informative without inclusion of the genetic factor.
Response: We sincerely appreciate the reviewer's valuable suggestion. Although this study did not include maternal secretor status as an independent variable in the multiple regression analysis. However, we conducted stratified analyses based on secretor types to further examine the genetic effects. Specifically, we separately analyzed the associations between metabolic factors and oligosaccharides in different secretor groups by multiple regression analysis. The detailed results presented in Tables S1 to 4. The results showed that the associations were different in the secretor and non-secretor mothers. In non-secretor mothers, 2′-FL was positively associated with maternal TC, LDL, and DBP. LNFP I was positively associated with maternal SBP and DBP. LNT and LNnT were positively associated with DBP (Supplementary Tables S1 and S2). Among secretor mothers, pre-pregnancy BMI was negatively correlated with 3′-SL, while maternal HR showed positive correlations with 3-FL, LNDFH II, and 3′-SL (Supplementary Tables S3 and S4).
- Please specify what are the numbers attached to the arrows in Fig. 4.
Response: We sincerely apologize for the incomplete description in manuscript. We have now added annotations for numerical labels on arrows in Figure 4.
The annotation was as follows:
The number above the arrow represented regression coefficient; * : P<0.05 for correlation coefficient.
We tried our best to improve the manuscript and made some changes in the manuscript. We appreciate for Editors and Reviewers’ warm work earnestly, and hope that the correction will meet with approval. Once again, thank you very much for your comments and suggestions.
Yours
Sincerely
Zhi Huang
Round 2
Reviewer 2 Report
Comments and Suggestions for Authors
I agree with the revisions made by the authors. The manuscript is now clear and well-structured. I consider it suitable for publication.
Author Response
Dear Reviewer
Thank you for your positive evaluation and acceptance of our manuscript. We sincerely appreciate your time and insightful comments throughout the review process, which have significantly improved the quality of our work.
Best regards,
Huang Zhi
Reviewer 3 Report
Comments and Suggestions for Authors
Inclusion of secretor/non-secrator in regression analysis will tell you the relative contribution of genetic and metabolic factors to oligosaccharide concs in human milk samples of this study.
According to the authors' response, the number attached to the arrow is a regression coefficient between the variables: is it standardized or not? Please specify.
Author Response
Dear Reviewer
We are very grateful for your advices and comments about our manuscript. Those were helpful to improve our paper. We had made a point-by-point revision and hope to meet with approval. Revised portion are marked in red in the paper.The responds to the reviewer’s comments are as following:
Inclusion of secretor/non-secrator in regression analysis will tell you the relative contribution of genetic and metabolic factors to oligosaccharide concs in human milk samples of this study.
Response:
We thank the reviewer for this valuable suggestion. Based on this feedback and the specific context of our study, we analyzed the associations between metabolic factors, secretor status, and concentrations of LNDFH II and 3′-SL using linear regression, with adjustments for maternal age and lactation period. As shown in Figure 4, maternal secretor status demonstrated a stronger positive association with LNDFH II levels compared to HR. However, no significant associations were observed between secretor status and 3′-SL concentrations.
Additionally, in the Discussion section, we have further expanded our commentary on these results as follows:
Additionally, the observed link was exclusive to secretor mothers. For LNDFH, maternal secretor status showed a stronger positive association with LNDFH II levels than HR. These findings suggested a potential role for maternal genetic factors. However, no significant associations were observed between secretor status and 3′-SL concentrations by linear regression analysis. The possible reasons may involve the relatively small sample size of non-secretor mothers in this study, or potential interaction effects between maternal secretor status and metabolic factors on 3′-SL concentrations.
According to the authors' response, the number attached to the arrow is a regression coefficient between the variables: is it standardized or not? Please specify.
Response: According to reviewer’s advice, we specify the the number attached to the arrow. The number above the arrow represented unstandardized regression coefficients.
We tried our best to improve the manuscript and made some changes in the manuscript. We appreciate for Editors and Reviewers’ warm work earnestly, and hope that the correction will meet with approval. Once again, thank you very much for your comments and suggestions.
Yours
Sincerely
Zhi Huang